

# Estimating global ammonia (NH$_3$) emissions based on IASI observations from 2008 to 2018

Zhenqi Luo[1, 2], Yuzhong Zhang[1, 2, *], Wei Chen[1, 2], Martin Van Damme[3, 4], Pierre-François Coheur[3], Lieven Clarisse[3]

[1]Key Laboratory of Coastal Environment and Resources of Zhejiang Province, School of Engineering, Westlake University, Hangzhou, Zhejiang Province, 310024, China
[2]Institute of Advanced Technology, Westlake Institute for Advanced Study, Hangzhou, Zhejiang Province, 310024, China
[3]Université libre de Bruxelles (ULB), Spectroscopy, Quantum Chemistry and Atmospheric Remote Sensing (SQUARES), Brussels, Belgium
[4]BIRA-IASB - Belgian Institute for Space Aeronomy, Brussels, Belgium

*Correspondence to*: Y. Zhang (zhangyuzhong@westlake.edu.cn), Z. Luo (zl725@cornell.edu)

**Abstract.** Emissions of ammonia (NH$_3$) to the atmosphere impact human health, climate, and ecosystems through their critical contributions to secondary aerosol formation. Estimation of NH$_3$ emissions is associated with large uncertainties because of inadequate knowledge about agricultural sources. Here, we use satellite observations from the Infrared Atmospheric Sounding Interferometer (IASI) and simulations from the GEOS-Chem model to constrain global NH$_3$ emissions over the period of 2008-2018. We update the prior NH$_3$ emission fluxes with the ratio between biases in simulated NH$_3$ concentrations and effective NH$_3$ lifetimes against the loss of the NH$_x$ family. In contrast to about a factor of two discrepancies between top-down and bottom-up emissions found in previous studies, our method results in a global land NH$_3$ emission of 79 (71-96) Tg a$^{-1}$, ~30 % higher than the bottom-up estimates. Regionally, we find that the bottom-up inventory underestimates NH$_3$ emissions over the South America and tropical Africa by 60-70 %, indicating under-representation of agricultural sources in these regions. We find a good agreement within 10 % between bottom-up and top-down estimates over the U.S., Europe and eastern China. Our results also show significant increases in NH$_3$ emissions over India (13 % decade$^{-1}$), tropical Africa (33 % decade$^{-1}$), and South America (18 % decade$^{-1}$) during our study period, consistent with the intensifying agricultural activities in these regions in the past decade. We find that inclusion of SO$_2$ column observed by satellite is crucial for more accurate inference of NH$_3$ emission trends over important source regions such as India and China where SO$_2$ emissions have changed rapidly in recent years.

## 1 Introduction

Emissions of ammonia (NH$_3$) to the atmosphere has critical implications for human health, climate, and ecosystems. As the main alkaline gas, NH$_3$ reacts with acidic products from precursors such as NO$_x$ and SO$_2$ to form fine particulate matters, which is a well-documented risk factor for human health, causing great welfare loss globally (Erisman 2021; Gu et al., 2021).



These particulate matters also affect the Earth's radiative balance by directly scattering incoming radiation (Ma et al., 2012)
and indirectly as cloud condensation nuclei (Höpfner et al., 2019). Additionally, both gas-phase ammonia ($NH_3$) and aerosol-
phase ammonium ($NH_4^+$) can deposit onto the surface of land and water through dry and wet processes, and therefore are
associated with soil acidification (Zhao et al., 2009), ecosystem eutrophication (Dirnböck et al., 2013), biodiversity loss
(Stevens et al., 2010), and cropland nitrogen uptake (Liu et al., 2013).
$NH_3$ is emitted from a variety of anthropogenic and natural sources, including agriculture, industry, fossil fuel combustion,
biomass burning, natural soils, ocean, and wild animals (Behera et al., 2013). Among these, agricultural activities, mainly
livestock manure management and mineral fertilizer application, are the most important $NH_3$ sources, which account for ~70%
of the total $NH_3$ emissions globally (Bouwman et al., 1997; Sutton et al., 2013). $NH_3$ emissions can be estimated with a
bottom-up approach based on information of emission activities and emission factors (Hoesly et al., 2018; Crippa et al.,
2021). However, bottom-up estimates of $NH_3$ emissions are generally thought to be uncertain, relative to other pollutants that
are mainly from fossil fuel combustion sources (e.g., $NO_x$, CO). One of the challenges is that the intensity of agricultural
$NH_3$ emissions, emission factors, either from livestock or fertilizer, depend strongly on management and farming practices,
but this information is usually not widely available (Zhang et al., 2017). Furthermore, microbial activities that are
responsible for agricultural $NH_3$ emissions are highly variable and has a complex dependence on environmental conditions,
which is often inadequately captured by bottom-up approaches (Behera et al., 2013; Vira et al., 2021). In many cases,
emission factors used in bottom-up modelling are based on local studies that are not representative for the diversity of
conditions and not depending on meteorological parameters.
Top-down analyses of atmospheric observations (e.g., $NH_3$ concentrations or $NH_4^+$ depositional fluxes) provide an alternative
constraint on $NH_3$ emissions. For example, observations of $NH_3$ concentrations and $NH_4^+$ deposition fluxes from surface
networks can be used to infer regional $NH_3$ emission fluxes (e.g., Paulot et al., 2014). However, surface sites are often sparse,
especially in developing continents such as Africa and South America, limiting our capability to constrain $NH_3$ emissions
globally. The advent of satellite observations makes it possible to investigate long-term spatially resolved $NH_3$ emissions
from national, continental, to global scales. Van Damme et al. (2018) reported large $NH_3$ point sources across the globe that
are detected by the Infrared Atmospheric Sounding Interferometer (IASI) instrument but missing in the bottom-up
inventories. Studies have also applied satellite data (e.g., IASI and Cross-track Infrared Sounder (CrIS)) to study $NH_3$
emissions from important source regions, including the U.S. (Cao et al., 2020; Chen et al., 2021b), China (Zhang et al.,
2018), and Europe (Marais et al., 2021; van der Graaf et al., 2021). These regional studies show 20 % to 50 % differences
between top-down and bottom-up estimates of $NH_3$ emissions.
Compared to regional analyses, long-term global analyses of $NH_3$ emissions based on satellite observations are relatively
scarce (e.g., Evangeliou et al., 2021). This is partly because of the computational challenges arising from a full-fledged
inversion for a long period of time and over large spatial extents. In a recent study, Evangeliou et al. (2021) proposed a fast
top-down method, in which $NH_3$ emissions are computed as the ratio between $NH_3$ column observations and $NH_3$ lifetime.
This method relies on $NH_3$ lifetime diagnosed from a chemical transport model (CTM) and assumes a local mass balance.



Their analysis found a global NH$_3$ emission of around 180 Tg a$^{-1}$, which is roughly triple the widely used bottom-up
estimates (e.g., 62 Tg a$^{-1}$ by the Community Emission Data System, CEDS). This large upward adjustment, if true, would
have huge implications for global reactive nitrogen cycles and indicate that our current understanding of global NH$_3$
emissions is seriously flawed.
In this paper, we examine if the large discrepancy between the bottom-up and top-down estimates is due to the methodology.
We refine the fast top-down approach by improving NH$_3$ lifetime diagnosis and partially accounting for the transport
contributions. We develop a series of data filtering procedures to exclude results that are not sufficiently constrained by
observations or affected by large deviations from the assumption of the fast top-down method. We apply the updated method
to IASI observations to derive the global distribution of NH$_3$ emissions fluxes from 2008 to 2018, and examine the impact of
the improved method on global NH$_3$ emission inferences. Finally, we evaluate the consistency of varied top-down and
bottom-up estimates against IASI observations with full-chemistry simulations.

## 78   2 Methods

### 79   2.1 IASI observations

We use 2008-2018 NH$_3$ total column retrievals (ANNI-NH$_3$-v3R) from the IASI on board Metop-A. The IASI instrument
measures the infrared radiation (645–2760 cm$^{-1}$) from Earth's surface and the atmosphere with a circular 12 km footprint at
nadir (Clerbaux et al., 2009; Van Damme et al., 2017). The retrieval algorithm calculates the hyperspectral range index from
IASI spectra measurements (Van Damme et al. 2014) and converts it to the NH$_3$ total column density via an artificial neural
network (Whitburn et al., 2016; Franco et al., 2018). The retrieval uses consistent meteorological data from the ERA5
reanalysis, so it is suitable for the analyses of inter-annual variability and long-term trends (Hersbach et al., 2020). The
ANNI-NH$_3$-v3R product, has been validated against in situ measurements and is shown to have a good regional correlation
(Guo et al., 2021; Van Damme et al., 2021). The dataset has been used in previous studies to estimate NH$_3$ emissions
globally (e.g., Evangeliou et al., 2021) and regionally (e.g., Chen et al., 2021b; Marais et al., 2021).
Here we only use morning NH$_3$ data (around 9:30 local solar time) though IASI provides global coverage twice daily,
because of the better precision of morning observations resulting from favorable thermal contrast conditions (Clarisse et al.
2010). We filter out data with a cloud fraction greater than 10 % (Van Damme et al., 2018) and a skin temperature below
263 K (Van Damme et al., 2014). The skin temperature dataset is from ERA5 (Hersbach et al., 2020). To compare with
simulated NH$_3$ columns (see **Sect. 2.2**), we regrid and average monthly IASI NH$_3$ observations on the GEOS-Chem 4° × 5°
grid (**Fig. 1a**). To reduce uncertainty from sparse sampling, we further exclude grid cells with the number of successful
retrievals less than 800 in a month. We also test the choices of the threshold for 400 and 1200 per month in the sensitivity
calculations (**Table S1**, line 5-6). This criterion affects mainly high latitudes during wintertime, where snow surfaces make it
unfavourable for infrared measurements (**Fig. S1**).

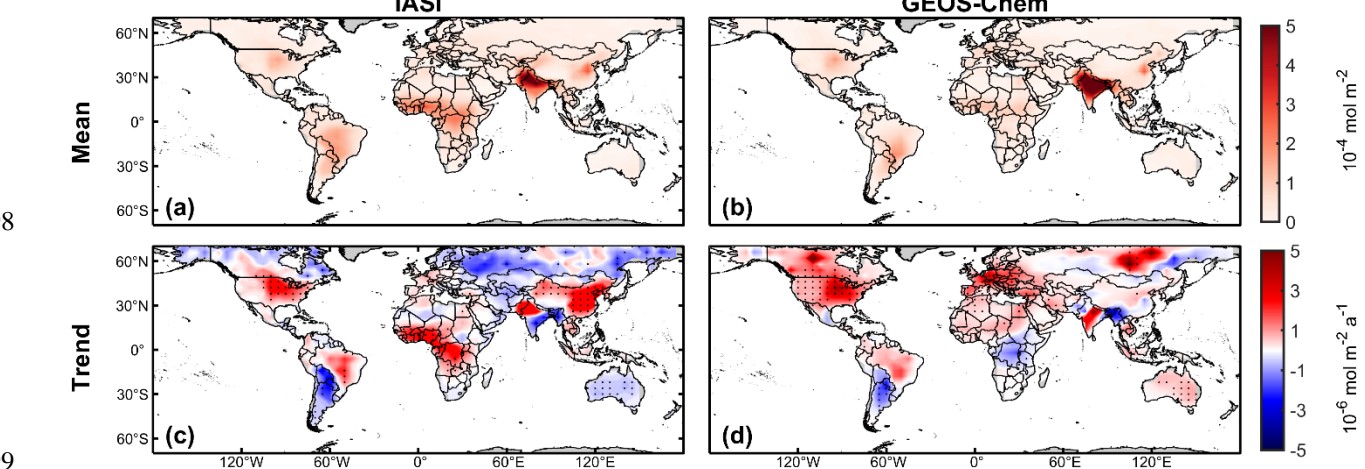

**Figure 1.** Spatial distribution of (a, c) IASI and (b, d) GEOS-Chem NH$_3$ column concentrations. (a, b) Mean and (c, d) linear trends within the **70**°N-**70**°S during 2008-2018. Dots in (c) and (d) indicate that linear trends are significant at the 95 % confidence levels. Linear trends are computed from the time series of annual averages.

## 2.2 GEOS-Chem simulations

We use the GEOS-Chem CTM v12.9.3 (10.5281/zenodo.3974569) to simulate global NH$_3$ concentrations. The GEOS-Chem model, driven by the MERRA-2 reanalyzed meteorology (Gelaro et al., 2017), simulates the tropospheric ozone–NOx–VOCs–aerosol chemistry at 4° × 5° resolution with 47 vertical layers (30 layers in the troposphere) (Bey et al., 2001; Park et al., 2004). The thermodynamic equilibrium between gas phase NH$_3$ and aerosol phase NH$_4^+$ is explicitly simulated by the ISORROPIA-II module in GEOS-Chem (Fountoukis & Nenes, 2007). The model also simulates the wet and dry deposition of NH$_3$ and NH$_4^+$, the terminal sinks of atmospheric NH$_x$ ($\equiv$ NH$_3$ + NH$_4^+$). Dry deposition is represented with a resistances-in-series scheme (Wesely, 2007) and wet deposition includes scavenging in convective updrafts and in- and below-cloud scavenging from large-scale precipitation (Wang et al., 2011; Amos et al., 2012). Anthropogenic emissions of simulated chemicals including those of NH$_3$ are taken from a global emission inventory CEDS (Hoesly et al., 2018), overridden by regional inventories in Canada (Air Pollutant Emission Inventory, APEI), the United States (2011 National Emissions Inventory, NEI-2011), Asia (MIX-Asia v1.1) (Li et al., 2017), and Africa (DICE-Africa) (Eloise Marais and Christine Wiedinmyer, 2016). Such compiled anthropogenic emissions only include incomplete information on inter-annual trends because inventories are not all available throughout the whole period. Anthropogenic emissions are essentially invariant after 2013 in our setup (**Fig. S2**). The general lack of trends in SO$_2$ emissions in the simulation, if not accounted for, may cause biases in inferred trends over regions such as India and China where SO$_2$ emissions have changed rapidly (Sun et al., 2018; Qu et al., 2019; Chen et al., 2021a). Fire emissions are from Global Fire Emissions Database (GFED4) (van der Werf et al., 2017), and biogenic VOC emissions are from the Model of Emissions of Gases and Aerosols from Nature (MEGAN) (Guenther et al., 2012). Temporal (seasonal and inter-annual) variations in fire and biogenic emissions are resolved by the inventories. Hereafter, we refer to NH$_3$ emissions from this set of inventories as BUE1. For comparison, we also use another





set    of    bottom-up    inventories    which    consist    of    EDGARv5.0    for    anthropogenic    emissions
(https://data.jrc.ec.europa.eu/collection/edgar, last access: 8 March 2022, Crippa et al., 2020), GFAS for fire emissions
(CAMS, https://apps.ecmwf.int/datasets/data/cams-gfas/, last access: 8 March 2022) (minor natural emissions are the same
as BUE1), which we denote as BUE2.
The GEOS-Chem simulation is conducted from 2008 to 2018 with an additional 1-month spin-up starting from December
2007. We sample the simulated $NH_3$ and $NH_4^+$ concentration fields between 9:00 to 10:00 local solar time, approximately the
IASI morning overpass time. To compare with the IASI $NH_3$ columns, we integrate the vertical profiles of simulated $NH_3$
concentrations by layer thickness. The ANNI-$NH_3$-v3R retrieval algorithm does not provide information on the vertical
sensitivity of the IASI measurements (i.e., averaging kernels) (Van Damme et al., 2017). In addition, we also archive
depositional and transport rates for $NH_3$ and $NH_4^+$, which are used in emission fluxes estimation.

### 2.3 NH₃ emission fluxes estimation

We compute the top-down $NH_3$ emission fluxes (TDE) ($\hat{E}_{NH_3}$, in molecules m$^{-2}$ s$^{-1}$) in land grid cells for individual months
from 2008 to 2018. We update the prior model emission fluxes ($E_{NH_3,mod}$, in molecules m$^{-2}$ s$^{-1}$) with a correction term
positively proportional to the difference of observed ($C_{NH_3,obs}$, in molecules m$^{-2}$) and simulated ($C_{NH_3,mod}$, in molecules m$^{-2}$)
$NH_3$ total column densities and inversely proportional to the lifetime of $NH_3$ ($\tau_{NH_3,mod}$, in s):

$$\hat{E}_{NH_3} = E_{NH_3,mod} + \frac{C_{NH_3,obs} - C_{NH_3,mod}}{\tau_{NH_3,mod}}, \tag{1}$$

where $\tau_{NH_3,mod}$ is computed as the ratio of the simulated $NH_3$ column and the sum of simulated loss rate of the $NH_x$ family
($NH_x \equiv NH_3 + NH_4^+$) through the dry and wet depositions of $NH_3$ ($D_{NH_3,mod}$, in molecules m$^{-2}$ s$^{-1}$) and $NH_4^+$ ($D_{NH_4^+,mod}$, in
molecules m$^{-2}$ s$^{-1}$):

$$\tau_{NH_3,mod} = \frac{C_{NH_3,mod}}{D_{NH_3,mod} + D_{NH_4^+,mod}}. \tag{2}$$

Here we consider the loss of the $NH_x$ family rather than that of $NH_3$, because the fast thermodynamic equilibrium between
gas-phase $NH_3$ and aerosol/aqueous-phase $NH_4^+$ implies that the conversion from $NH_3$ to $NH_4^+$ is not a terminal loss for $NH_3$
from the atmosphere. The $NH_3$ lifetime may be underestimated over source regions and overestimated over remote regions,
if $NH_3$ to $NH_4^+$ conversions are treated as a terminal loss as in Evangeliou et al. (2021) rather than a partition within a
chemical family as in **Eq. (2)**.
In addition, our method linearizes the column-emission relationship at prior emissions as opposed to zero emissions in the
previous method (e.g., Evangeliou et al., 2021). Here, the baseline $NH_3$ column ($C_{NH_3,mod}$) simulated by the GEOS-Chem
model explicitly accounts for the non-local contribution of transport, while the correction to prior emissions is done only
locally, that is, the difference between $C_{NH_3,obs}$ and $C_{NH_3,mod}$ is attributed only to errors in local emissions without
accounting for the sensitivity to emissions from other grid cells. This hybrid approach can partially include the non-local



contribution from transport but still keeps the computation tractable for a long-term study such as this study, striking a trade-
off between the computational efficiency of a local method (e.g., Van Damme et al., 2018; Evangeliou et al., 2021) and the
accuracy of a full-fledged inversion (e.g., Cao et al., 2020; Chen et al., 2021b). The errors arising from local correction of
$NH_3$ emissions are expected to be small in most cases, because the $NH_3$ lifetime is short relative to a typical transport time
across a $4° \times 5°$ grid cell on which emissions are estimated. To identify cases when this error is not negligible, we apply a
monthly $NH_x$ budget analysis based on the GEOS-Chem simulation and exclude grid cells from our analysis where transport
dominates over local prior emissions or depositions in the monthly $NH_3$ budget (Transport/Emission>1 or
Transport/Deposition>1). We also test the impact of alternative thresholds (0.2 and 5) on $NH_3$ emission estimations (**Table
S1**, Line 7-8). This procedure mostly affects remote regions where emissions are small, notably northern high latitudes (**Fig.
S3**).
Because rapid changes in $SO_2$ emissions in eastern China and India, particularly after 2012, are not captured by our prior
simulation (**Fig. S2**), the estimation of $NH_3$ emission trends using **Eq. (1)** may be biased over these regions. To address this
issue, we further modify **Eq. (1)** to include observed trends in $SO_2$ column concentrations:

$$\hat{E}_{NH_3,SO_2-correct} = E_{NH_3,mod} + \frac{C_{NH_3,obs} - C_{NH_3,mod} + 2\omega C_{SO_4^{2-},mod}}{\tau_{NH_3,mod}},\qquad(3)$$

where $\omega$ (%) is the fractional changes of average $SO_2$ columns relative to the baseline year (i.e., 2012) over China or India
and $C_{SO_4^{2-},mod}$ (molecules m$^{-2}$ s$^{-1}$) is the simulated column densities of aerosol sulfate. Here, we specify a linear trend of -5 %
yr$^{-1}$ for eastern China and 5 % yr$^{-1}$ for India between 2012 and 2018, based on values derived from the ozone monitoring
instrument (OMI) and Ozone Mapping and Profiler Suite (OMPS) observations (Wang and Wang, 2020; Liu et al., 2018).
The factor 2 accounts for the fact that two molecules of $NH_3$ are required to neutralize one molecule of $H_2SO_4$. **Eq. (3)** only
applies when $NH_3$ is in excess, a condition usually met in eastern China and India but not necessarily elsewhere (Lachatre et
al., 2019; Acharja et al., 2022). Therefore, we only apply **Eq. (3)** to eastern China and India to understand the impact of
changing $SO_2$ emissions on the inference of $NH_3$ emission trends. To use $SO_2$ observations systematically in $NH_3$ emission
estimations requires further investigations.

### 2.4 Uncertainty and sensitivity analysis

We perform a series of perturbation and sensitivity experiments to assess the uncertainty of our estimates (**Table S1**). We
perturb $C_{NH_3,mod}$ and $\tau_{NH_3,mod}$ in **Eq. (1)**. The perturbations to $\tau_{NH_3,mod}$ are set to be 50 % and 200 % (**Table S1**, Line 1-2).
The perturbation to $C_{NH_3,mod}$ is set to be the standard deviation of monthly mean column concentrations ($\sigma_{C,obs}$) (**Table S1**,
Line 3-4), which is related to the number of IASI measurements ($n$) and their measurement errors:

$$\sigma_{C,obs} = \sqrt{\frac{\sum_{i=1}^{i=n}(\sigma_i \times \Omega_i)^2}{n-1}},\qquad(4)$$



where $\Omega_i$ (in mol m$^{-2}$) is the $i^{th}$ NH$_3$ column measurement out of a total number of $n$ observations in a grid cell during a
month and $\sigma_i$ is the relative error reported in the IASI product. We then use $\Omega \pm \sigma_{C,obs}$ to evaluate the effect of measurement
errors in emission estimates (**Table S1**, Line 3-4). We also conduct sensitivity tests by using alternative parameters in data
filtering (**Table S1**, Line 5-8).
In addition, we perform GEOS-Chem full chemistry simulations in selected years (2008, 2013, 2018) to examine the
consistency of NH$_3$ emission estimates with the IASI observations. We use our top-down estimate (TDE) and prior
emissions (BUE1) to drive the full chemistry simulation. We compute the fractional biases (FBs) of these simulations against
the IASI observations to evaluate the systematic biases in the resulting NH$_3$ column density fields:
$$FB = 2 \times \frac{\sum_{i=1}^{i=N}(C_{mod,i} - C_{obs,i})}{\sum_{i=1}^{i=N}(C_{mod,i} + C_{obs,i})}. \tag{5}$$

**3 Results and discussion**
**3.1 Observed and simulated NH$_3$ concentrations**
**Fig. 1a and 1b** plot observed and simulated NH$_3$ total column concentrations averaged over 2008-2018. The GEOS-Chem
simulation generally reproduces the global distribution of NH$_3$ concentrations observed by the IASI instrument. Good
agreements (i.e., difference < 10 %) are found in the U.S., Europe, and southern South America. Meanwhile, the GEOS-
Chem model underestimates NH$_3$ concentrations in eastern China, northern South America, and tropical Africa by 20-120 %,
and overestimates in southern India by around 50 %, indicating biases in NH$_3$ emissions over these regions.
**Fig. 1c and 1d** show 2008-2018 linear trends in NH$_3$ column concentrations derived from the IASI observations and the
GEOS-Chem simulations. The linear trends are computed based on the time series of annual averages. The IASI trends
shown in Fig. 1c are in general consistent with a recent analysis by Van Damme et al. (2021). IASI observes a positive NH$_3$
concentration trend of 2.9 % a$^{-1}$ over the U.S., and this trend is well captured by GEOS-Chem. Similarly, the observation and
the simulation agree on a dipole pattern in South America (i.e., positive trend in Brazil and negative trend in Argentina).
Because anthropogenic emissions over this region are set to be invariant in our simulation (**Fig. S2**), this agreement suggests
that these trends are due to meteorological conditions and/or fire emissions, rather than changes in anthropogenic emissions.
The satellite also observes significant positive trends in NH$_3$ concentrations over China (5.2 % a$^{-1}$) and tropical Africa (2.0 %
a$^{-1}$), but these trends are not reproduced in the simulation (0.3 % a$^{-1}$ for China and 0.2 % a$^{-1}$ for tropical Africa). These
simulation-observation differences can not only reflect discrepancies in the trends of anthropogenic NH$_3$ emissions, but also
be attributed to uncaptured changes in SO$_2$ and/or NO$_x$ emissions in these regions. We also find that a positive NH$_3$
concentration trend over Europe appears in the simulation (3.0 % a$^{-1}$) but is much weaker (1.0 % a$^{-1}$) in the observation,
suggesting decreasing emissions after 2013. Both the satellite and model do not find significant trends in NH$_3$ concentrations
over India (absolute value less than 1 % yr$^{-1}$). Strong GEOS-Chem trends in eastern Canada and Siberia result from large





wildfires that occurred in the latter part of the study period. IASI trends in northern boreal regions are less robust because of
noisy and sparse measurements over high latitudes (**Fig. S1**).

**3.2 NH₃ emissions inferred from IASI observations**

**Fig. 2** shows the spatial distributions of NH₃ emission fluxes and their 2008–2018 linear trends inferred from IASI
observations using the method described in **Sect. 2.3**. **Fig. 3** plots annual time series aggregated for seven selected regions.
The top-down estimate (TDE) suggests upward adjustments in NH₃ emissions over South America (SA) by 62 %, tropical
Africa (TA) by 69 %, and Central Asia (CA) by 327 %, relative to the prior inventory (BUE1), but downward adjustments in
NH₃ emissions by 14 % in India Peninsula (IP) and by 33 % in Canada. After accounting for the contributions from natural
emissions including fires, we find that most of these biases in NH₃ emissions can be attributed to anthropogenic sources,
except for Canada where the underestimation appears to relate to fire emissions. This result reflects a general inadequate
representation of agricultural and industrial emissions from developing continents in current global emission inventories.
The TDE finds good agreements with the BUE1 (difference within 10 %) over the U.S., Europe (EU), eastern China (EC)
and Australia.

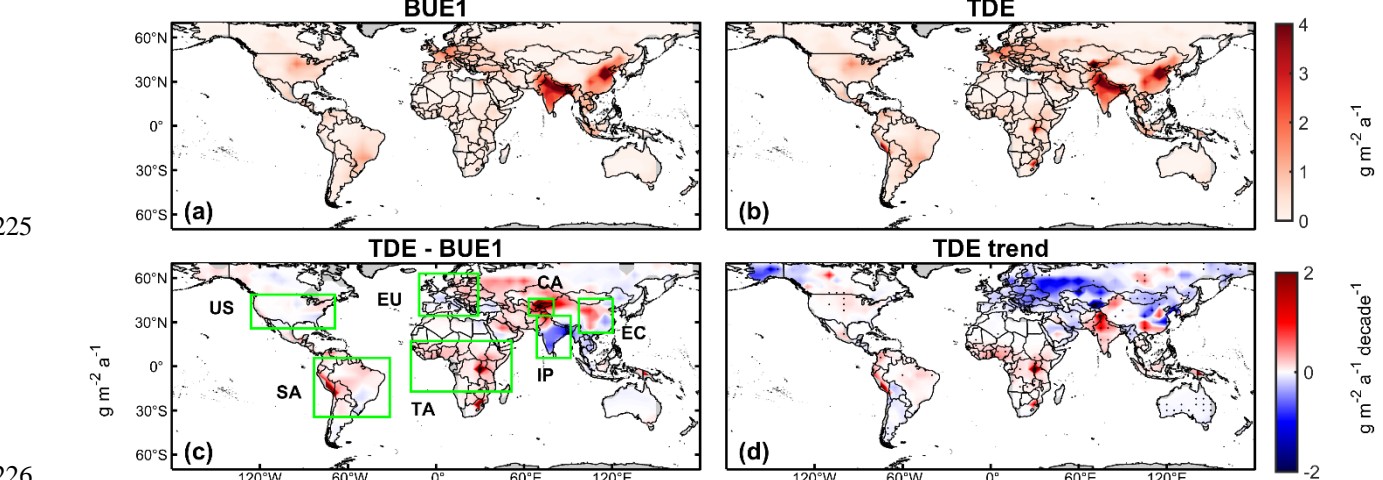

**Figure 2.** Spatial distribution of NH₃ emission fluxes during 2008-2018. (a) Bottom-up emissions (BUE1), (b) top-down emissions (TDE) inferred from IASI observations, (c) difference between TDE and BUE1 estimates and (d) emission trends derived from TDE estimates. Green boxes denote seven regions analysed in Sect. 3.2. Top-down emission fluxes are computed with Eq. (1) except for IP and EC where Eq. (3) is applied. Linear trends are computed from the time series of annual averages. Dots in (d) represent significant linear trends at the 95 % confidence level.

In addition to the adjustments in average emissions, the TDE also detects changes in NH₃ emissions during the period of
2008-2018, as expressed in linear trends computed from annual time series. We find significant positive emission trends in
SA (1.7 Tg a⁻¹ decade⁻¹ or 18 % decade⁻¹) and TA (2.8 Tg a⁻¹ decade⁻¹ or 33 % decade⁻¹) (**Fig. 3**). These increases are
concurrent with intensifying agricultural activities in these regions (Warner et al., 2017; E. Hickman et al., 2020), except for
a 2010 peak over SA, which coincides with fires in savanna and evergreen forests there (Chen et al., 2013). Comparison with
data from the Food and Agriculture Organization of the United Nations (FAO) (http://www.fao.org/faostat, last access: 7
July 2021) suggests that the increase in SA is driven primarily by growing application of synthetic fertilizer (55 % decade$^{-1}$),
whereas the increase in TA is consistent with increasing manure amount (28 % decade$^{-1}$) from a growing livestock
population (E. Hickman et al., 2021) (**Fig. 4**).

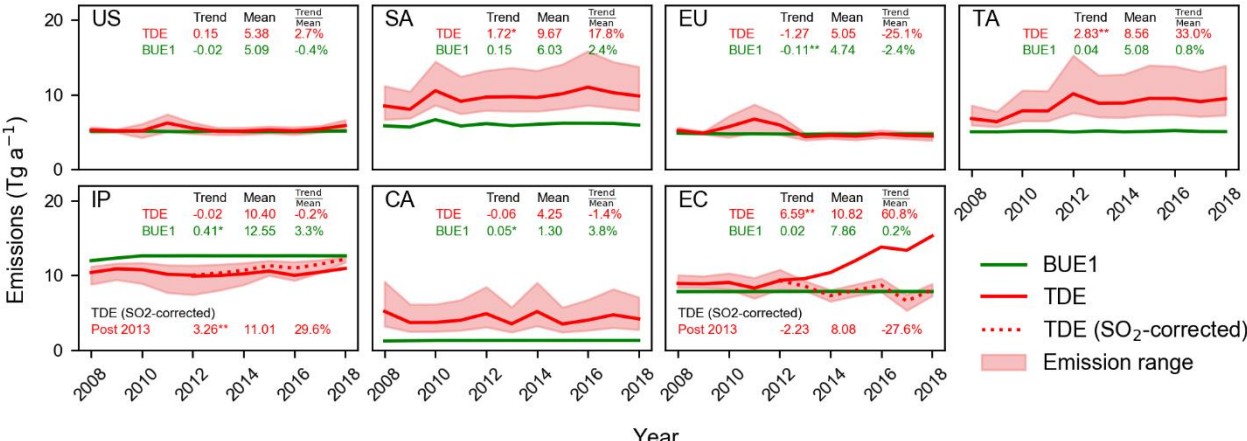


**Figure 3.** Annual NH$_3$ emissions for seven selected regions during 2008-2018. Shadings represent the range derived from uncertainty
analyses (see Sect. 2.4). Average annual emissions (Tg a$^{-1}$), absolute linear trends (Tg a$^{-1}$ decade$^{-1}$) and relative trends (% decade$^{-1}$) for
2008-2018 are inset. The asterisk symbols '*' and '**' represent that linear trends are significant at the 95 % and 99 % confidence level,
respectively. Red dashed lines represent top-down NH$_3$ emission estimates over IP and EC during 2013-2018, based on Eq. (3) that
accounts for observed trends of SO$_2$ (denoted as "SO$_2$-corrected"). Statistics for this estimate are also inset.

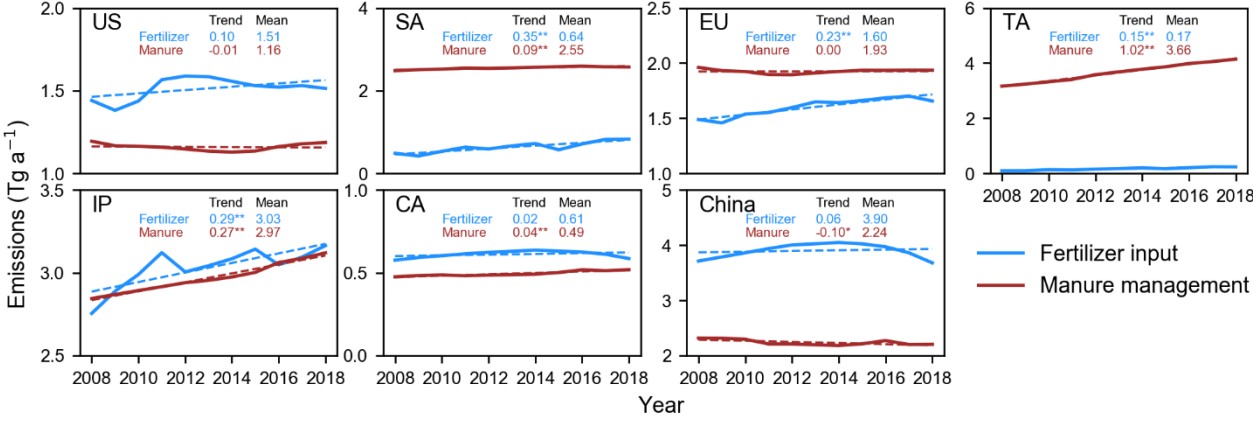


**Figure 4.** Synthetic fertilizer and manure management based on FAO reports (http://www.fao.org/faostat) during 2008-2018. To roughly
compare the contribution from the two sectors, we convert FAO reported statistics to NH$_3$ emissions (Tg a$^{-1}$) by applying fixed emission
factors of 13 % for manure N contents (Ma et al., 2020) and 17 % for synthetic fertilizer N contents (Riddick et al., 2016). Values of
means (Tg a$^{-1}$) and linear trends (Tg a$^{-1}$ decade$^{-1}$) are inset. Scales differ between panels.
Our results infer large but variable trends over northern high latitudes (e.g., negative trends in Alaska, central Russia, and
eastern Europe, but positive trends in Canada) (**Fig. 2d**). Because of large uncertainties associated with high-latitude



observations and emission optimization, these trends are less robust but can be partly attributed to variations in fire activities.
Decreases in Russia and eastern Europe are related to wildfire of boreal forests in early part of the study period (2008-2011)
(Keywood et al., 2012; Warner et al., 2017), while emission increases in Canada is due to wildfire in the late part of the
period (2015) (Pavlovic et al., 2016). We also infer negative trends (-43 % decade$^{-1}$) in Australia, which are statistically
significant, but the absolute magnitude of these trends is small (-0.03 g m$^{-2}$ a$^{-1}$ decade$^{-1}$ in **Fig. 2d**). The TDE estimation does
not find significant trends in NH$_3$ total emissions over the US and Central Asia.

**3.3 Impact of changing SO$_2$ emissions on NH$_3$ emission trends over eastern China and India**

Based on only NH$_3$ column measurements (**Eq. (1)**), we also find a decadal increase of 61 % decade$^{-1}$ (6.6 Tg a$^{-1}$ decade$^{-1}$) in
NH$_3$ emissions over eastern China (**Fig. 3**). This increase is especially large after 2013 and is driven mainly by increases of
IASI NH$_3$ column concentration in eastern China (**Fig. 1c**). This large post-2013 increase is inconsistent with flat or even
declining fertilizer input and manure amount (**Fig. 4**). On the other hand, we find no appreciable emission trend in IP (**Fig. 3**),
which appears to agree with relatively stable IASI NH$_3$ concentrations over the period (**Fig. 1c**) but is not supported by
increases in fertilizer applications and manure amount shown in the FAO report (**Fig. 4**).
An assumption underlying **Eq. (1)** is that the model simulation captures the partition between gas-phase NH$_3$ and aerosol-
phase NH$_4^+$. In addition to alkaline NH$_3$, the partition is also determined by the abundance of acids (e.g., H$_2$SO$_4$ and HNO$_3$).
Inaccurate emissions of their precursors (e.g., SO$_2$ and NO$_2$) in the model simulation, in particular over regions with
excessive NH$_3$, can lead to biases in simulating the NH$_3$-NH$_4^+$ partition. It is well known that SO$_2$ emissions in China have
decreased rapidly after 2013 because of stringent air pollution control measures (Sun et al., 2018; Zhai et al., 2021), while
SO$_2$ emissions from India have been increasing (Qu et al., 2019). But these regional trends are not captured in our prior
simulation due to a lack of emission data (**Fig. S2**).
We find that the discrepancies between top-down (**Eq. 1**) and bottom-up estimates of emission trends over EC and IP can be
largely reconciled by including observed SO$_2$ column concentrations in the top-down calculation (**Eq. (3)**). By accounting
for OMI and OMPS observed SO$_2$ trends (Wang and Wang, 2020), we derive an overall decreasing NH$_3$ emissions in EC
between 2013 and 2018 (-2.2 Tg a$^{-1}$ decade$^{-1}$, -28 % decade$^{-1}$). This result suggests that observed increases in NH$_3$ columns
over China are largely explained by decreases in SO$_2$ emissions (**Fig. 1** and **Fig. 3**), consistent with previous studies (Fu et
al., 2017; Liu et al., 2018; Lachatre et al., 2019; Chen et al., 2021a). Bottom-up inventories (e.g., MEIC v1.3, EDGAR v5.0)
also report stable or declining NH$_3$ emissions from China during the period (Li et al., 2017; Crippa et al., 2020). Meanwhile,
the revised method (**Eq. (3)**) finds a positive post-2013 trend (3.3 Tg a$^{-1}$ decade$^{-1}$, 30 % yr$^{-1}$) in NH$_3$ emissions over India.
Compared with our original estimate using **Eq. (1)**, NH$_3$ emission trends derived with **Eq. (3)** (i.e., decrease in China and
increase in India after 2013) is more consistent with the bottom-up information of fertilizer input and manure management
(**Fig. 4**). This result demonstrates the potential of assimilating both NH$_3$ and SO$_2$ satellite observations in constraining NH$_3$
emissions, which should be further explored in the future.





**3.4 Sensitivity of global emission inference to NH$_3$ lifetime diagnosis**

Integrating over land areas globally, our IASI-based TDE estimates of NH$_3$ emissions is 79 (71-96) Tg a$^{-1}$ (range of estimates from uncertainty analysis, see **Table S1**) (**Fig. 5**). This result is about 20-40 % higher than bottom-up inventories (BUE1, 62 Tg a$^{-1}$ and BUE2, 56 Tg a$^{-1}$). In contrast, a previous study by Evangeliou et al. (2021) also based on the IASI data estimated a much higher global NH$_3$ emission of 180 Tg a$^{-1}$ (**Fig. 5**). One cause of the difference between the two IASI-based estimates is in diagnosis of NH$_3$ lifetime from CTM. Evangeliou et al. (2021) treats conversion from NH$_3$ to NH$_4^+$ as a terminal loss and diagnoses NH$_3$ lifetime averaged 11.6 ± 0.6 h globally from a CTM, which is close to a constant NH$_3$ lifetime (12 h) assumed in Van Damme et al. (2018). In this study, we account for the fact that fast thermodynamic equilibrium can establish between NH$_3$ and NH$_4^+$ so that NH$_3$ can only be terminally lost through the deposition of the NH$_x$ family (**Eq. (2)**), which yields a global averaged NH$_3$ lifetime of 21.2 ± 3.8 h (**Fig. 6**). This longer NH$_3$ lifetime implies a higher sensitivity of NH$_3$ column density to NH$_3$ emissions, leading to a lower estimate for global NH$_3$ emissions. In addition, instead of locally scaling observed NH$_3$ column by lifetime (Van Damme et al., 2018; Evangeliou et al, 2021; Marais et al., 2021), our method (**Eq. (1)**) partially accounts for the non-local contribution from transport by including prior NH$_3$ columns from a full 3-D simulation and using their difference from observed NH$_3$ columns to correct prior emissions, which prevents derivation of large NH$_3$ emissions in remote regions where observed NH$_3$ concentrations are driven mainly by transport. Our data filtering strategy (**Sect 2.1 and 2.2**) is also crucial to avoid spurious top-down results when satellite coverage is poor and the local mass balance assumption does not hold.

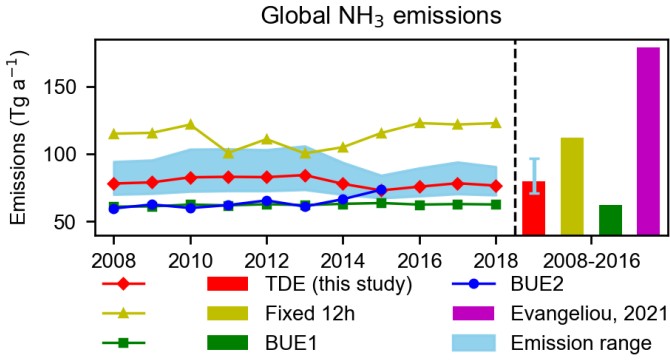

**Figure 5.** Comparison of our top-down NH$_3$ emission estimates (TDE) with other top-down (Fixed 12h and Evangeliou et al. (2021)) and bottom-up (BUE1 and BUE2) results during 2008-2018. The red line and red bar represent central estimates of the TDE, and the blue shaded area and the blue error bar indicate the uncertainty evaluated by our study (**Sect. 2.4**).

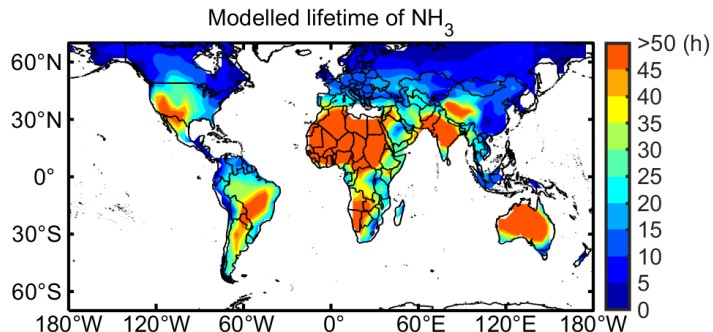

**Figure 6.** Spatial distribution of NH₃ lifetime (h) diagnosed from GEOS-Chem (Eq. (2)) within the **70°N-70°S** during 2008-2018.

**Fig. 6** shows the spatial variation in NH₃ lifetime diagnosed from the GEOS-Chem simulation. Short NH₃ lifetimes (< 10 h) are found mainly in northern high latitudes. Short lifetime in eastern China is due to high wet NH₄⁺ deposition velocity, although some regional studies suggested an overestimation of deposition fluxes by the model especially in forest areas (e.g., Yangtze River basin) (Zhao et al., 2017; Xu et al., 2018). Very long NH₃ lifetime (> 100 h) occurs over Sahara and Australia, where dry conditions result in slow wet deposition.

We then evaluate the consistency of NH₃ emissions derived from varied methods with IASI observations using full GEOS-Chem simulations in the selected years of 2008, 2013, and 2018. Results are shown in **Fig. 7** (fractional bias, FB) and **Table S2** (number of valid grid cells, $R^2$, and root mean square error). The full-chemistry GEOS-Chem simulations driven the prior emissions (BUE1) tends to underestimate NH₃ column density (mean FB ~-30%), while that driven by our TDE emission estimates with improved NH₃ lifetime calculation achieves better consistency with observations (mean FB ~10%). The fact that the TDE is more consistent with IASI observations demonstrates the superiority of the improved top-down method.

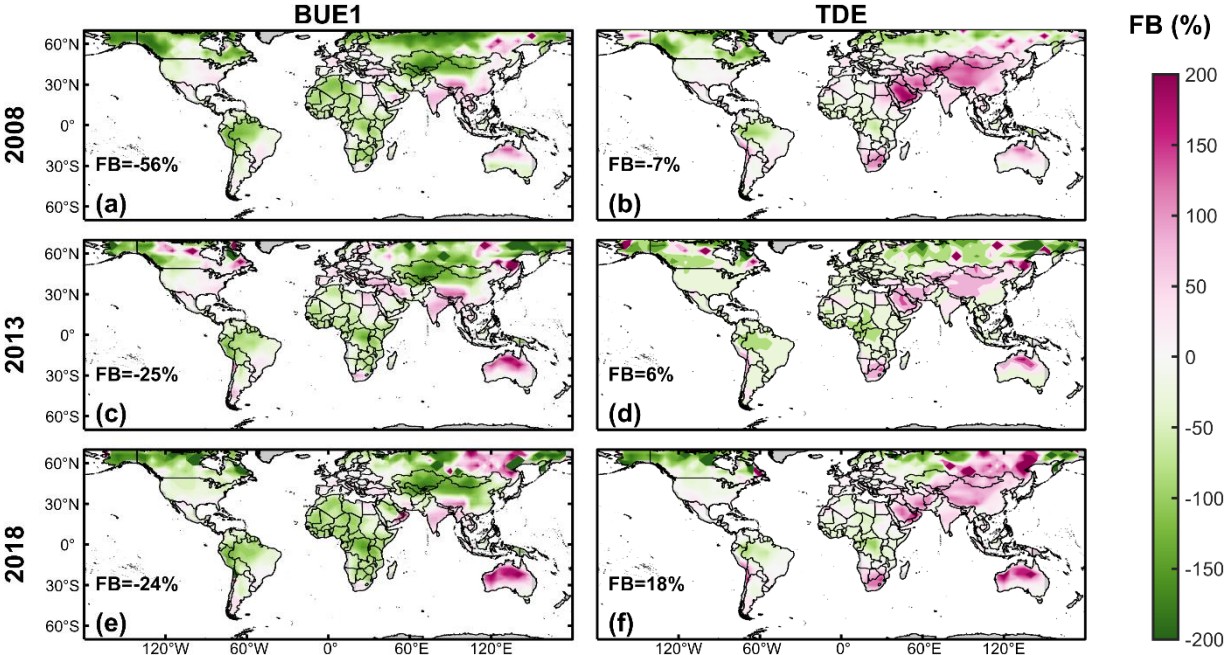

**Figure 7.** Fractional biases of simulated NH$_3$ column densities from GEOS-Chem simulations driven by (a, c, e) BUE1 and (b, d, f) TDE for the year (a-b) 2008, (d-e) 2013 and (g-h) 2018, against IASI observations. Global average FBs (%) for each year are inset.

## 4 Conclusions

This study quantifies global ammonia (NH$_3$) fluxes monthly from 2008 to 2018 at 4° × 5° resolution, through a fast top-down method that incorporates IASI satellite observations and GEOS-Chem model simulations. The top-down method updates the prior NH$_3$ emissions with a correction term positively proportional to the difference of the observed and simulated NH$_3$ concentrations, and inversely proportional to the lifetime diagnosed from a CTM. This method revises previously proposed fast top-down methods in two aspects. First, we account for thermodynamic equilibrium within the NH$_x$ family in diagnosing NH$_3$ lifetime, while previous studies either assume a globally constant lifetime or treat conversion from NH$_3$ to NH$_4^+$ as a terminal sink. Second, our formulation linearizes the column-emission relationship at prior emissions as opposed to zero emissions in the previous method, which in general reduces errors from the local mass balance approximation. Another improvement is that we apply several data filtering procedures to exclude unreliable top-down results that are not sufficiently constrained by observations or affected by large deviations from the local mass balance assumption.

We apply this improved fast top-down method to IASI NH$_3$ column observations from 2008 to 2018. We find that the bottom-up inventory (BUE1) underestimates NH$_3$ emission over South America (62 %) and tropical Africa (69 %), but overestimates over India (14 %) and Canada (33 %). The bottom-up inventory agrees with the top-down estimate over the U.S., Europe, and eastern China (i.e., within 10 %). Our analysis also shows significant increases in India (13 % decade$^{-1}$),





tropical Africa (33 % decade$^{-1}$), and South America (18 % decade$^{-1}$) during the study period, consistent with intensifying
agricultural activities over these regions. An analysis of agricultural statistics suggests that the increase in tropical Africa is
likely driven by growing livestock population and that in South America by increasing fertilizer usage.
We show that large increases in $NH_3$ concentrations in eastern China is mainly driven by rapid decreases in $SO_2$ emissions in
recent years. By accounting for observed $SO_2$ columns, we find that $NH_3$ emissions from eastern China is significantly
decreasing during 2008-2018 (-19 % decade$^{-1}$), with a larger negative trend after 2013 (-28 % decade$^{-1}$), as compared to a
significant positive trend (61 % decade$^{-1}$) derived from assimilating only $NH_3$ data. Similarly, a lack of trend in observed
$NH_3$ concentrations over India is due to concurrent increases in $SO_2$ and $NH_3$ emissions. After including observed $SO_2$
columns in the calculation, we estimate a 13 % increase in $NH_3$ emissions over India, with a significant post-2013 positive
trend (30 % decade$^{-1}$). These results from assimilating both $NH_3$ and $SO_2$ data is more consistent with the agricultural
statistics in China and India.
Our estimate for global total $NH_3$ emission is 79 (71-96) Tg a$^{-1}$, about 30 % higher than the BUE1 estimate. This contrasts
with a much higher estimate (180 Tg a$^{-1}$) derived from Evangeliou et al. (2021) also using IASI data. The discrepancy can be
primarily attributed to a longer $NH_3$ lifetime (i.e., global average 21 h) diagnosed in our method, which represents a greater
sensitivity of $NH_3$ column to emissions, and a more conservative data filtering strategy, which removes potentially unreliable
top-down results. Our diagnosis of $NH_3$ lifetime is an improvement over Evangeliou et al. (2021), by accounting for the
thermodynamic equilibrium between gas phase $NH_3$ and aerosol phase $NH_4^+$ in our formula. We show with full chemistry
simulations, our top-down estimate achieves better consistency with IASI observations, compared to the bottom-up emission
inventory (BUE1).

*Data availability.*
The IASI L2 ammonia satellite observations are available at the AERIS data infrastructure (https://iasi.aeris-data.fr/). The
ERA5 skin temperature and GFAS fire emission can be request through Copernicus Climate Data Store
(https://cds.climate.copernicus.eu/cdsapp#!/home). Agricultural data are available through Food and Agriculture
Organization of the United Nations (FAO) (http://www.fao.org/faostat). The GEOS-Chem model can be retrieved from
10.5281/zenodo.3974569. All the other data and scripts used for the present publication can be obtained from the
corresponding author upon request.
*Author contributions.*
ZL and YZ designed the study. ZL performed the simulations and analyses and wrote and coordinated the paper. WC
contributed to the model simulations for consistency evaluation. LC, MVD, and PFC developed the IASI-NH3 satellite
product. ZL and YZ wrote the paper with inputs from all authors.



*Acknowledgements.*
This study is supported by Westlake University. We thank the High-Performance Computing Center of Westlake University
for the facility support and technical assistance. We acknowledge the AERIS data infrastructure https://www.aeris-data.fr for
providing access to the IASI data. The IASI L1c data are received through the EUMETCast near real-time data distribution
service. Research at ULB was supported by the Belgian State Federal Office for Scientific, Technical and Cultural Affairs
(Prodex HIRS) and the Air Liquide Foundation (TAPIR project). LC is Research Associate supported by the Belgian F.R.S.-
FNRS. Hersbach et al., (2020) was downloaded from the Copernicus Climate Change Service (C3S) Climate Data Store. The
results contain modified Copernicus Climate Change Service information 2020. Neither the European Commission nor
ECMWF is responsible for any use that may be made of the Copernicus information or data it contains. IASI is a join
mission of Eumetsat and the "Centre National d'Études Spatiales" (CNES, France). We acknowledge the constructive
comments and suggestions from Prof. Peter Hess from the Cornell University and Dr. Yi Wang from the University of Iowa.
We also acknowledge Dr. Nikolaos Evangeliou from Norwegian Institute for Air Research for providing his $NH_3$ emission
flux data and for discussions with ZL.

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
