# Peer review of "Estimating global ammonia (NH3) emissions based on IASI observations from 2008 to 2018"

_Atmospheric Chemistry and Physics, 2022_

## Author Comment (AC1)

**Responses to the Manuscript ACP-2022-216: Estimating global ammonia (NH₃) emissions based on IASI observations from 2008 to 2018**

Dear Editor-in-Chief:

We hereby submit the revised version of our manuscript (ACP-2022-216).

We greatly appreciate you and two referees for providing highly insightful and constructive comments, which have substantially improved the clarify of our manuscript. We have carefully addressed all these comments, please see below our point-to-point **responses in blue** and **red** text and refer to the **revised manuscript**.

In our attached documents, we have made the following major revisions:

- Add a section about the uncertainty evaluation and add two figures in this section.
- Move original Table S1 to the main text and add more explanations on the uncertainty and sensitivity analysis.
- Move original Figure 7 to the supplementary and add several figures in the supplementary.

We hope you find our manuscript suitable for publication and look forward to hearing from you

Yours sincerely, Zhenqi Luo, Yuzhong Zhang, Wei Chen, Martin van Damme, Pierre-François Coheur, Lieven Clarisse

Email: zhangyuzhong@westlake.edu.cn, zl725@cornell.edu

**1. RESPONSES TO REFEREE, REFEREE #1**

**Comment #1: General comments**

This manuscript showcases global NH₃ emissions estimated using a top-down approach constrained by IASI observations with comparison to bottom-up estimates. The approach used here is built upon a previous study with modifications in several aspects including NH₃ lifetime calculation and local mass balance approximation. The authors address the uncertainty of emissions by performing sensitivity tests on various parameters and discuss the limitations of current emission inventories. This work shows the promise of using satellite observations to constrain NH₃ emission inventories on the global scale, as well as the need to improve emission factors used in model simulations, particularly in developing regions. Overall, the manuscript is well-written and organized. The figures are clear, concise, and easy to follow. I recommend the publication of this manuscript in ACP with some minor comments/suggestions for the authors to address/consider.

We thank the referee for positive evaluation of the manuscript.

**Comment #2: Specific comments**

Line 80: You may want to emphasize this is the reanalyzed IASI dataset as opposed to the near real time dataset.

We add the text in **Line 79**.

Line 89-97: You may also want to state explicitly that you only used observations taken over land areas.

We add the text in **Line 92 and 136**.

Line 103 (Sect 2.2): One major distinction between this study and Evangeliou et al. (2021), besides modifications in the approach, is the CTM used for simulating $NH_3$. Can you elaborate a little on the differences between GEOS-Chem and LMDz-OR-INCA, and any outcomes they may have on the emission estimates?

The systematic comparison is out the scope of our study, and we do not find reference indicating that model differences are the major cause of discrepancies in emission inference.

Line 253-257: Are these speculations or do you have statistical evidence to support? Without seeing the inter-annual variabilities of emissions or some analyses on the $NH_3$/CO ratio, I am not completely sure if the positive trend in Canada can be explained by one particular wildfire season, likewise the negative trend in Russia and eastern Europe.

The fire emission inventory (Global Fire Emissions Database, GFED4, van der Werf et al., 2017) shows that large fires occurred in Canada are from 2008 to 2011 and in eastern Europe are from 2013 to 2016 and also 2017 (**Fig. S4**, now added in the Supplement). We attribute trends to fire events mainly based on this inventory. We now clarify in the text (**Line 253**).

[Figure]

**Figure S4.** Monthly average of $NH_3$ fire emission over Canada (130°-50°W, 48°-84°N) and eastern Europe (10°-55°E, 36°-72°N) during 2008-2018. The emission data is from GFED4.

Line 282-285: I wonder how good the bottom-up estimates are in India and China, as you mentioned that emission factors in developing regions may not be as accurate. In Fig. 3, BUE1 in IP and EC is almost invariant throughout the whole period, which seems contradictory with the fertilizer and manure data in Fig. 4. The reason for asking this is even though the $SO_2$ correction largely closes the gap between BUE1 and TDE, if prior emissions are off in the first place do we have enough confidence to agree on the absolute magnitude of the emissions?

We now clarify here and elsewhere (e.g., caption of **Fig. 3**; **Sect. 2.2**) that the prior inventory implemented in our simulation has incomplete annual information on emissions (including $NH_3$ and $SO_2$) especially after 2013, and that's why it is inconsistent with fertilizer and manure data. We use CEDS as the default anthropogenic inventory replaced by MIX-Asia v1.1 in Asia. The objective of including both observed $NH_3$ and $SO_2$ in $NH_3$ emission quantification is to reduce dependence on prior emissions (of $NH_3$ and $SO_2$).

Line 332: What percentage of results or grid cells were determined as unreliable and removed?

Supplement: Table 2 should be renamed as Table S2. Also and again, I think showing the percentage of grids used may be more meaningful than just the number of grids.

We applied a monthly $NH_x$ budget analysis based on the GEOS-Chem simulation and exclude grid cells from our analysis where transport dominates over local prior emissions or depositions in the monthly $NH_3$ budget. The total percentage of excluding grid cells range from 0 to 40 % for the seven selected regions. We now show this information in **Fig. S6** and **Table S1**.

[Figure]

**Figure S6.** Total percentages of excluding grid cells for seven selected regions between 2008-2018.

Finally, the novelty of this study and why it is important should be highlighted more given the overlap with Evangeliou et al. (2021) in terms of topic, datasets, and methods. I was hoping to see a more conclusive statement on the implication of this

work to the scientific community interested in $NH_3$ emissions. You modified the fast top-down approach proposed by Evangeliou et al. (2021), and the resulting change in emission estimates is drastic (79 vs 180 Tg a$^{-1}$). My takeaway from this is the emission figures one will get from models are largely subject to the method they choose and assumptions they make. Do you think this approach is scientifically reasonable enough for the purpose of deriving global emissions, or is a full-fledged inversion still necessary to obtain more accurate numbers? And how might the results change if a different satellite product is used (CrIS, for example)?

Our method is particularly useful for long-term global analysis of emission trends (or changes). It is better than direct trend analysis of $NH_3$ column density (as often applied in current literature), because our method handles the effect of meteorology. In addition, including observed $SO_2$ accounts for the impact of $SO_2$ on $NH_3/NH_4^+$ partition, which is even conceptually better than a full-fledged inversion that only assimilate $NH_3$ observations. However, this is only applied in China and India in this study, as more investigation is needed to extend the idea globally because of variations in the regime of sulfate-nitrate-ammonium aerosol. We have modified the conclusion to include these points.

**Comment #3: Technical corrections**

Line 30: Emissions of ammonia ($NH_3$) to the atmosphere have…

Line 31: such as nitrogen oxides ($NO_x$) and sulfur dioxide ($SO_2$)

Line 276: an overall decreasing trend in $NH_3$ emissions

Line 316: driven by the prior emissions

Line 343: $NH_3$ emissions from eastern China are significantly decreasing

References: I see a few citations messed up due to subscripts in the titles (e.g., line 477, line 496 and 497). Please correct accordingly.

We have corrected them. Thank you!

**2. RESPONSES TO REFEREE, REFEREE #2**

**Comment #1: General**

Ammonia emissions estimates constructed from bottom-up inventories currently have large uncertainties. Top-down estimates of ammonia emissions using ammonia retrievals from satellite-borne instruments have the potential to greatly refine these bottom-up estimates. This paper uses ammonia retrievals from IASI to construct a global top-down estimate of ammonia emissions. The authors build on the work of Evangeliou et al., 2021, offering several improvements. These new emissions estimates yield much fewer emissions globally than derived in of Evangeliou et al., 2021, but significantly larger than the bottom-up estimates.

While a number of studies have recently been conducted that use similar methods to derive ammonia emissions regionally, currently few studies have provided top-down ammonia emission estimates globally. As such, this is a timely and scientifically relevant paper. Overall, the presentation of this paper is good, but some re-organization should be considered to improve the paper's readability. Additional analysis is also required before publication, as outlined below.

Thank you for reviewing our paper and for your helpful suggestions.

**Comment #2: Major comments**

I could not see any comparisons with an independent set of observations (such as surface observations). Validation with another set of observations needs be added. For this, calculations of FB, $R^2$, and RMSE should be examined (as done in Table S2).

We add the comparison of simulated concentrations with ground-based measurements, i.e., validation against the ground-based measurements (**Line 339-351**), though available observations are not sufficient to evaluate our major findings in regions like tropical Africa, South America, and South Asia.

Section 2.1 should contain some discussion of the uncertainties associated with the IASI $NH_3$ retrievals. There is a reference to using the relative errors that are reported with the retrievals in Section 2.4, so maybe part of line 183 could be moved to Section 2.1? What are typical uncertainties on the IASI retrievals?

We add the discussion of uncertainty on the IASI retrievals in terms of its vertical profile (**Line 128-131**).

The information related to the sensitivity analysis seems out of order, as this information is spread out over different sections. Also, by referring to Table S1 in Section 2.1 and 2.4, it seems to present results of the emissions inversion before the main results are presented. I would move the sentences starting with 'To reduce uncertainty' on line 94 to the end of line 97 to either Section 2.3 or 2.4. I would consider moving the sentence starting with 'We also test' on line 160 to Section 2.4 as well. Also, I think Table S1 should be moved from the Supplement to the main body of the paper. I also think it would be better to move the second paragraph in Section 2.4 that start on line 186 to the end of Section 2.3. I cannot see much discussion of the results of the sensitivity analysis in the 'Results and discussion' section. More discussion of the sensitivity analysis should be included. In addition to discussing the range of results of the global emissions from the sensitivity analysis, it would be good to add results/discussion/figures for the sensitivity analysis for emissions on regional scales. For example, would it be possible to add a plot like Fig. 2b, but for it to show the deviation from the TDE results instead (maybe as a percentage of the TDE emissions) when the parameters in Table S1 are varied?

We have reorganized the description on sensitivity analysis in **Sect. 2.4**. We move **Table S1** to the main text, which summarizes all sensitivity tests. We also add more discussion in **Sect. 3.5** and **Fig. 7** on uncertainty analysis (**Line 324-331**).

[Figure]

**Figure 7**. Spatial distribution of TDE relative uncertainty as the discrepancy of emission estimations in parameters perturbation (Table 1) divided by the TDE average during 2008-2018.

At the end of Section 2.2, you mention that averaging kernels are not provided in the retrieval product used. Including the averaging kernels in the calculation of the columns could potentially significantly change the column values. Is there reason to think the effect of the averaging kernel on the column is small? If so, could you provide some details on this? If not, is it possible to make a rough estimation of the averaging kernel and see how much this changes the results? Also, the two sentences starting with 'To compare' on line 129 until the reference to Van Damme et al. 2017 on line 131 should be moved elsewhere (maybe to Section 2.3 or a new subsection with the new information requested).

We add more clarification in the text (**Line 128-131**).

More discussion of the rational for using the formulation in Eq. (1) is necessary. Different mass balance methods use different methods to determine the proportionality constant between the columns and emissions. For instance, in the finite difference mass balance (FDMB) method, this proportionality is determined by comparing two different model runs: one run using the a priori emissions and one run with perturbed emissions. In the FDMB method, if you assume that you have a simplified model that assumes a steady state and no transport, the proportionality constant will be 1/tau, as in Eqn. (1). However, in general these two different methods will differ. So I'm curious what the rational in choosing this method over the FDMD method is. Do you expect that the two methods would give similar results? Alternatively, an inversion/assimilation method could be used instead, where the proportionality constant would instead be given by the Kalman gain, which takes into account the uncertainties of both the observations and the a priori emissions estimates. Could you describe the advantages and/or disadvantages of using the Kalman gain instead of a mass balance method? In this context, I'm assuming the Kalman gain would be a scalar just like tau, not a (large) matrix that would be employed in a Kalman Filter method such as an EnKF that would obviously be very computationally expensive.

Yes, our method is different from the FDMB. In FDMB, the proportionality constant $\beta$ is calculated by perturbing emissions by a fixed amount and calculating the resulting change to the column abundance, which accounts for the sensitivity of

fractional changes in local columns to fractional changes in emissions if assumes a steady state and no transport. We suppose the difference of the emission estimation resulting from these two methods depend on the difference in $\beta$ and $\tau$ applied in the equations. The ensemble Kalman filter (EnKF) update the state estimate with the Kalman gain. It is computationally expensive for CTMs with dozens of model simulations so that it may not be suitable for the long-term global analysis like this study. The comparison of different mass balance methods is out the scope of our study.

On line 143, you mention that you use the lifetime for $NH_x$ instead of $NH_3$. The lifetimes of the two will differ, but I didn't quite follow why you would want to use the lifetime of NHx if the estimation is for the emissions of $NH_3$. Could you add more explanation why this makes sense conceptually?

We use lifetime of $NH_3$ (not $NH_x$) against the loss of the $NH_x$ family. We now add more explanations on why we think this calculation is proper. See **Line 145-149**.

In regards to Eqn. (3), if the concern is that $SO_2$ emissions are underestimated in the bottom-up emissions, and you would like to make a correction for this, why not just increase the bottom-up emissions estimates by the same amount (i.e. omega)? Is the method used easier to implement? If the increase in $SO_2$ emissions are fed into GEOS-Chem, then ISORROPIA-II can work out the details of the $NH_3$ - $NH_4^+$ partitioning, i.e when $NH_3$ is in excess, etc...

Eq. (3) combines information from both $NH_3$ and $SO_2$ observations for $NH_3$ emission quantification, because the two are intrinsically connected. The method reduces our dependence on accurate prior emissions of either $SO_2$ or $NH_3$. It also allows to correct the $SO_2$ effect, without rerun the model with corrected $SO_2$ emissions. So it should be easier to implement than the approach described by the referee.

One line 186, you mention that 'we perform GEOS-Chem full chemistry simulations in selected years'. What was the GEOS-Chem mode run initially for the full 2008-2018 period? This should be moved to Section 2.2 and expanded on, i.e. give a brief description of the simulation modes and their difference, why you chose to use the full chemistry mode for only these three years, etc... Also, with the way the paper is currently organized, it is a bit confusing figuring out which simulation mode was used for what. I assume that the non-full-chemistry mode was used with Eqns. (1) and (3), but I can't see where this is specified in the paper. Please clarify in Section 2.2 exactly what the output of each simulation mode is used for. A table might make this clearer to the reader.

Sorry for the confused organization. All the GEOS-Chem simulations we run are in the full chemistry mode. We have corrected it as the 'GEOS-Chem simulation'.

In regards to the calculation of fractional bias of IASI as compared to GEOS-Chem, if the IASI observations were already used in the estimates of the emissions through comparisons to GEOS-Chem (Eqns. (1) and (3)), what is the goal of presenting these FB values? This comparison uses the full chemistry mode (I assumed that Eqns. (1)

and (3) where used in non-full chemistry mode, is that correct?), but I'm not clear on what this comparison is telling you. Add more explanation of what this is being used for. If comparisons with an independent observation data set are added, as suggested above, is this comparison still necessary?

We now add more clarification in the text (**Line 332-338**). This exercise is to check whether results from Eq. (1) and (3) (which are a simplified description of $NH_3$ chemistry and transport) indeed improve agreements with IASI.

On line 263, it says that the 'large post-2013 increase is inconsistent with flat or even declining fertilizer input and manure amount (Fig. 4).' Do the plots in Fig. 4 include emissions from livestock? If not, it may be hard to really tell if the trends found by using Eqn. (1) are really inconsistent with the data in Fig. 4.

We now clarify that livestock emissions are related to livestock manure amount (**Fig. 4 and Line 265**).

On line 275, it described that $SO_2$ trends from OMI and OMPS are used with Eqn (3). The estimates of $SO_2$ trends from OMI and OMPS will have uncertainties associated with them. Could these uncertainties be incorporated into your analysis? i.e. could you include another sensitivity test for the value of the $SO_2$ trend used?

We report the suggested sensitivity test in **Line171**. Yes, Wang and Wang (2020) reported the stand deviation of the yearly $SO_2$ trend values as 10-20 %. Thus, we perturb the $\omega$ in the Eq. (3) as 4/6 % and -4/-6 % over China and India in our uncertainty analysis (**Line 187-188**).

**Comment #3: Minor comments**

Line 31: 'particulate matters' to 'particulate matter'

Line 33: 'These particulate matters also affect' to 'Particulate matter also affects'
Line 35: '… and wet processes, and therefore are associated with …' remove the word 'therefore'

Line 50: 'depending' should be changed to 'dependent'

Line 136: Would be good to remind readers here that C_{NH3,obs} are the monthly mean observations, not an individual observation.

Line 180: Replace 'related to the number of IASI measurements (ð• ' › ) and their measurement errors:' by 'given by'

Line 187: 'top-down estimate (TDE) and prior emissions (BUE1)' the acronym TDE was already defined on line 134, while the acronym BUE was used prior to this line but not defined until this line. Change so that these acronyms are defined at their first appearance in the paper.

Section 3.2 starts by introducing Fig. 2 and then introduces Fig. 3 in the next sentence. Try to reorganize this section by first discussing Fig. 2 and then move on to discussing Fig. 3 instead of introducing them at the start of the section together.

We correct them, thank you for the careful read.

In regards to Fig. 2, I'm a bit confused about the colour bars and the colour scale. Fig 2c shows the difference between Fig 2a and 2b, so the units should be in g m^{-2} a^{-1}, but Fig 2c has blue in it, which isn't in the colour bar in the top row that has these units. The bottom row colour bar has blue in it, but it has the wrong units for 2c (g m^{-2} a^{-1} per decade). Could you clarify which colour bar goes with Fig 2c? Could the figure be rearrange or changed in some way to make this more clear?

The units for **Fig. 2c** are g m$^{-2}$ a$^{-1}$ and we add a color bar for **Fig. 2c.**

[Figure]

**Figure 2.** Spatial distribution of NH3 emission fluxes during 2008-2018. (a) Bottom-up emissions (BUE1), (b) top-down emissions (TDE) inferred from IASI observations, (c) difference between TDE and BUE1 estimates and (d) emission trends derived from TDE estimates. Green boxes denote seven regions analyzed in Sect. 3.2. Top-down emission fluxes are computed with Eq. (1) except for IP and EC where Eq. (3) is applied. Linear trends are computed from the time series of annual averages. Dots in (d) represent significant linear trends at the 95 % confidence level.

In the description of Fig 3, it says 'Shadings represent the range derived from uncertainty analyses' Table S1 describes a number of different sensitivity tests. Does the shading refer to the upper and lower bounds over all tests? Please specify in the main text.

We add the description of the uncertainty analysis as the referee suggested (**Line 185-195**)

Line 210: 'Both the satellite and model do not find significant trends in NH$_3$ concentrations over India (absolute value less than 1 % yr$^{-1}$).' Looking at Figs. 1(c) and (d), it looks like the trend over India are not insignificant. Could you confirm this?

Thank you for your suggestion. The overall observed $NH_3$ concentration trend over India are -0.8 % $a^{-1}$ (p = 0.25) while trends in south-eastern part are significantly decreasing ($\sim$ -3.8 % $a^{-1}$, p < 0.05). The model does not find significant trends in $NH_3$ concentrations, with p > 0.1 in all grids. We revised the text accordingly (**Line 215-217**).

[Figure]

**Figure A.** Spatial distribution of relative linear trend of (a) IASI and (b) GEOS-Chem $NH_3$ column concentrations in India. Dots indicate that linear trends are significant at the 95 % confidence levels (p value < 0.05). Linear trends are computed from the time series of annual averages.

Line 261: Why is Eqn. (1) 'based only on NH3 column measurements' if Eqn. (1) blends information from the observations with a priori information of the emissions?

We correct following the suggestion (**Line 270**)

Line 295: Is the lifetime of 21.2 +/- 3.8 h the lifetime of NH3 (as stated) or is it the lifetime of NHx?

It is the lifetime of $NH_3$ as stated.

In Table S1, 'TDE' isn't really a 'parameter perturbed'. This line should probably be something like 'None' or 'None (TDE)'.

We correct **Table 1** following the suggestion.

**REFERENCE**

Evangeliou, N., Balkanski, Y., Eckhardt, S., Cozic, A., Van Damme, M., Coheur, P.-F., Clarisse, L., Shephard, M. W., Cady-Pereira, K. E., and Hauglustaine, D.: 10-year satellite-constrained fluxes of ammonia improve performance of chemistry

transport models, Atmospheric Chemistry and Physics, 21, 4431-4451, 10.5194/acp-21-4431-2021, 2021.

Van Damme, M., Whitburn, S., Clarisse, L., Clerbaux, C., Hurtmans, D., and Coheur, P.-F.: Version 2 of the IASI NH3; neural network retrieval algorithm: near-real-time and reanalysed datasets, Atmospheric Measurement Techniques, 10, 4905-4914, 10.5194/amt-10-4905-2017, 2017.

Van Damme, M., Clarisse, L., Whitburn, S., Hadji-Lazaro, J., Hurtmans, D., Clerbaux, C., and Coheur, P. F.: Industrial and agricultural ammonia point sources exposed, Nature, 564, 99-103, 10.1038/s41586-018-0747-1, 2018.

van der Werf, G. R., Randerson, J. T., Giglio, L., van Leeuwen, T. T., Chen, Y., Rogers, B. M., Mu, M., van Marle, M. J. E., Morton, D. C., Collatz, G. J., Yokelson, R. J., and Kasibhatla, P. S.: Global fire emissions estimates during 1997–2016, Earth System Science Data, 9, 697-720, 10.5194/essd-9-697-2017, 2017.

Wang, Y. and Wang, J.: Tropospheric SO2 and NO2 in 2012–2018: Contrasting views of two sensors (OMI and OMPS) from space, Atmospheric Environment, 223, 10.1016/j.atmosenv.2019.117214, 2020.

---

## Author Response (AR3)

**Responses to the Manuscript ACP-2022-216: Estimating global ammonia (NH₃) emissions based on IASI observations from 2008 to 2018**

Zhenqi Luo, Yuzhong Zhang, Wei Chen, Martin van Damme, Pierre-François Coheur, Lieven Clarisse

Email: zhangyuzhong@westlake.edu.cn, zl725@cornell.edu

**1. RESPONSES TO REFEREE, REFEREE #1 (Major Revision)**

**Comment #1: General comments**

This manuscript showcases global $NH_3$ emissions estimated using a top-down approach constrained by IASI observations with comparison to bottom-up estimates. The approach used here is built upon a previous study with modifications in several aspects including $NH_3$ lifetime calculation and local mass balance approximation. The authors address the uncertainty of emissions by performing sensitivity tests on various parameters and discuss the limitations of current emission inventories. This work shows the promise of using satellite observations to constrain $NH_3$ emission inventories on the global scale, as well as the need to improve emission factors used in model simulations, particularly in developing regions. Overall, the manuscript is well-written and organized. The figures are clear, concise, and easy to follow. I recommend the publication of this manuscript in ACP with some minor comments/suggestions for the authors to address/consider.

We thank the referee for positive evaluation of the manuscript.

**Comment #2: Specific comments**

Line 80: You may want to emphasize this is the reanalyzed IASI dataset as opposed to the near real time dataset.

We add the text in **Line 79**.

Line 89-97: You may also want to state explicitly that you only used observations taken over land areas.

We add the text in **Line 92 and 136**.

Line 103 (Sect 2.2): One major distinction between this study and Evangeliou et al. (2021), besides modifications in the approach, is the CTM used for simulating $NH_3$. Can you elaborate a little on the differences between GEOS-Chem and LMDz-OR-INCA, and any outcomes they may have on the emission estimates?

The systematic comparison is out the scope of our study, and we do not find reference indicating that model differences are the major cause of discrepancies in emission inference.

Line 253-257: Are these speculations or do you have statistical evidence to support? Without seeing the inter-annual variabilities of emissions or some analyses on the $NH_3/CO$ ratio, I am not completely sure if the positive trend in Canada can be explained by one particular wildfire season, likewise the negative trend in Russia and eastern Europe.

The fire emission inventory (Global Fire Emissions Database, GFED4, van der Werf et al., 2017) shows that large fires occurred in Canada are from 2008 to 2011 and in eastern Europe are from 2013 to 2016 and also 2017 (**Fig. S4**, now added in the Supplement). We attribute trends to fire events mainly based on this inventory. We now clarify in the text (**Line 253**).

[Figure]

**Figure S4.** Monthly average of $NH_3$ fire emission over Canada (130°-50°W, 48°-84°N) and eastern Europe (10°-55°E, 36°-72°N) during 2008-2018. The emission data is from GFED4.

Line 282-285: I wonder how good the bottom-up estimates are in India and China, as you mentioned that emission factors in developing regions may not be as accurate. In Fig. 3, BUE1 in IP and EC is almost invariant throughout the whole period, which seems contradictory with the fertilizer and manure data in Fig. 4. The reason for asking this is even though the $SO_2$ correction largely closes the gap between BUE1 and TDE, if prior emissions are off in the first place do we have enough confidence to agree on the absolute magnitude of the emissions?

We now clarify here and elsewhere (e.g., caption of **Fig. 3**; **Sect. 2.2**) that the prior inventory implemented in our simulation has incomplete annual information on emissions (including $NH_3$ and $SO_2$) especially after 2013, and that's why it is inconsistent with fertilizer and manure data. We use CEDS as the default anthropogenic inventory replaced by MIX-Asia v1.1 in Asia. The objective of including both observed $NH_3$ and $SO_2$ in $NH_3$ emission quantification is to reduce dependence on prior emissions (of $NH_3$ and $SO_2$).

Line 332: What percentage of results or grid cells were determined as unreliable and removed?

Supplement: Table 2 should be renamed as Table S2. Also and again, I think showing the percentage of grids used may be more meaningful than just the number of grids.

We applied a monthly $NH_x$ budget analysis based on the GEOS-Chem simulation and exclude grid cells from our analysis where transport dominates over local prior emissions or depositions in the monthly $NH_3$ budget. The total percentage of excluding grid cells range from 0 to 40 % for the seven selected regions. We now show this information in **Fig. S6** and **Table S1**.

[Figure]

**Figure S6.** Total percentages of excluding grid cells for seven selected regions between 2008-2018.

Finally, the novelty of this study and why it is important should be highlighted more given the overlap with Evangeliou et al. (2021) in terms of topic, datasets, and methods. I was hoping to see a more conclusive statement on the implication of this work to the scientific community interested in $NH_3$ emissions. You modified the fast top-down approach proposed by Evangeliou et al. (2021), and the resulting change in emission estimates is drastic (79 vs 180 Tg a$^{-1}$). My takeaway from this is the emission figures one will get from models are largely subject to the method they choose and assumptions they make. Do you think this approach is scientifically reasonable enough for the purpose of deriving global emissions, or is a full-fledged inversion still necessary to obtain more accurate numbers? And how might the results change if a different satellite product is used (CrIS, for example)?

Our method is particularly useful for long-term global analysis of emission trends (or changes). It is better than direct trend analysis of $NH_3$ column density (as often applied in current literature), because our method handles the effect of meteorology. In addition, including observed $SO_2$ accounts for the impact of $SO_2$ on $NH_3/NH_4^+$ partition, which is even conceptually better than a full-fledged inversion that only assimilate $NH_3$ observations. However, this is only applied in China and India in this study, as more investigation is needed to extend the idea globally because of variations in the regime of sulfate-nitrate-ammonium aerosol. We have modified the conclusion to include these points.

**Comment #3: Technical corrections**

Line 30: Emissions of ammonia ($NH_3$) to the atmosphere have...

Line 31: such as nitrogen oxides ($NO_x$) and sulfur dioxide ($SO_2$)

Line 276: an overall decreasing trend in $NH_3$ emissions

Line 316: driven by the prior emissions

Line 343: $NH_3$ emissions from eastern China are significantly decreasing

References: I see a few citations messed up due to subscripts in the titles (e.g., line 477, line 496 and 497). Please correct accordingly.

We have corrected them. Thank you!

**2. RESPONSES TO REFEREE, REFEREE #2 (Major Revision)**

**Comment #1: General**

Ammonia emissions estimates constructed from bottom-up inventories currently have large uncertainties. Top-down estimates of ammonia emissions using ammonia retrievals from satellite-borne instruments have the potential to greatly refine these bottom-up estimates. This paper uses ammonia retrievals from IASI to construct a global top-down estimate of ammonia emissions. The authors build on the work of Evangeliou et al., 2021, offering several improvements. These new emissions estimates yield much fewer emissions globally than derived in of Evangeliou et al., 2021, but significantly larger than the bottom-up estimates.

While a number of studies have recently been conducted that use similar methods to derive ammonia emissions regionally, currently few studies have provided top-down ammonia emission estimates globally. As such, this is a timely and scientifically relevant paper. Overall, the presentation of this paper is good, but some re-organization should be considered to improve the paper's readability. Additional analysis is also required before publication, as outlined below.

Thank you for reviewing our paper and for your helpful suggestions.

**Comment #2: Major comments**

I could not see any comparisons with an independent set of observations (such as surface observations). Validation with another set of observations needs be added. For this, calculations of FB, $R^2$, and RMSE should be examined (as done in Table S2).

We add the comparison of simulated concentrations with ground-based measurements, i.e., validation against the ground-based measurements (**Line 339-351**), though available observations are not sufficient to evaluate our major findings in regions like tropical Africa, South America, and South Asia.

Section 2.1 should contain some discussion of the uncertainties associated with the IASI NH$_3$ retrievals. There is a reference to using the relative errors that are reported with the retrievals in Section 2.4, so maybe part of line 183 could be moved to Section 2.1? What are typical uncertainties on the IASI retrievals?

We add the discussion of uncertainty on the IASI retrievals in terms of its vertical profile (**Line 128-131**).

The information related to the sensitivity analysis seems out of order, as this information is spread out over different sections. Also, by referring to Table S1 in Section 2.1 and 2.4, it seems to present results of the emissions inversion before the main results are presented. I would move the sentences starting with 'To reduce uncertainty' on line 94 to the end of line 97 to either Section 2.3 or 2.4. I would consider moving the sentence starting with 'We also test' on line 160 to Section 2.4 as well. Also, I think Table S1 should be moved from the Supplement to the main body of the paper. I also think it would be better to move the second paragraph in Section 2.4 that start on line 186 to the end of Section 2.3. I cannot see much discussion of the results of the sensitivity analysis in the 'Results and discussion' section. More discussion of the sensitivity analysis should be included. In addition to discussing the range of results of the global emissions from the sensitivity analysis, it would be good to add results/discussion/figures for the sensitivity analysis for emissions on regional scales. For example, would it be possible to add a plot like Fig. 2b, but for it to show the deviation from the TDE results instead (maybe as a percentage of the TDE emissions) when the parameters in Table S1 are varied?

We have reorganized the description on sensitivity analysis in **Sect. 2.4**. We move **Table S1** to the main text, which summarizes all sensitivity tests. We also add more discussion in **Sect. 3.5** and **Fig. 7** on uncertainty analysis (**Line 324-331**).

[Figure]

**Figure 7**. Spatial distribution of TDE relative uncertainty as the discrepancy of emission estimations in parameters perturbation (Table 1) divided by the TDE average during 2008-2018.

At the end of Section 2.2, you mention that averaging kernels are not provided in the retrieval product used. Including the averaging kernels in the calculation of the columns could potentially significantly change the column values. Is there reason to think the effect of the averaging kernel on the column is small? If so, could you provide some details on this? If not, is it possible to make a rough estimation of the

averaging kernel and see how much this changes the results? Also, the two sentences starting with 'To compare' on line 129 until the reference to Van Damme et al. 2017 on line 131 should be moved elsewhere (maybe to Section 2.3 or a new subsection with the new information requested).

We add more clarification in the text (**Line 128-131**).

More discussion of the rational for using the formulation in Eq. (1) is necessary. Different mass balance methods use different methods to determine the proportionality constant between the columns and emissions. For instance, in the finite difference mass balance (FDMB) method, this proportionality is determined by comparing two different model runs: one run using the a priori emissions and one run with perturbed emissions. In the FDMB method, if you assume that you have a simplified model that assumes a steady state and no transport, the proportionality constant will be 1/tau, as in Eqn. (1). However, in general these two different methods will differ. So I'm curious what the rational in choosing this method over the FDMD method is. Do you expect that the two methods would give similar results? Alternatively, an inversion/assimilation method could be used instead, where the proportionality constant would instead be given by the Kalman gain, which takes into account the uncertainties of both the observations and the a priori emissions estimates. Could you describe the advantages and/or disadvantages of using the Kalman gain instead of a mass balance method? In this context, I'm assuming the Kalman gain would be a scalar just like tau, not a (large) matrix that would be employed in a Kalman Filter method such as an EnKF that would obviously be very computationally expensive.

Yes, our method is different from the FDMB. In FDMB, the proportionality constant $\beta$ is calculated by perturbing emissions by a fixed amount and calculating the resulting change to the column abundance, which accounts for the sensitivity of fractional changes in local columns to fractional changes in emissions if assumes a steady state and no transport. We suppose the difference of the emission estimation resulting from these two methods depend on the difference in $\beta$ and $\tau$ applied in the equations. The ensemble Kalman filter (EnKF) update the state estimate with the Kalman gain. It is computationally expensive for CTMs with dozens of model simulations so that it may not be suitable for the long-term global analysis like this study. The comparison of different mass balance methods is out the scope of our study.

On line 143, you mention that you use the lifetime for $NH_x$ instead of $NH_3$. The lifetimes of the two will differ, but I didn't quite follow why you would want to use the lifetime of NHx if the estimation is for the emissions of $NH_3$. Could you add more explanation why this makes sense conceptually?

We use lifetime of $NH_3$ (not $NH_x$) against the loss of the $NH_x$ family. We now add more explanations on why we think this calculation is proper. See **Line 145-149**.

In regards to Eqn. (3), if the concern is that $SO_2$ emissions are underestimated in the bottom-up emissions, and you would like to make a correction for this, why not just

increase the bottom-up emissions estimates by the same amount (i.e. omega)? Is the method used easier to implement? If the increase in $SO_2$ emissions are fed into GEOS-Chem, then ISORROPIA-II can work out the details of the $NH_3$ - $NH_4^+$ partitioning, i.e when $NH_3$ is in excess, etc...

Eq. (3) combines information from both $NH_3$ and $SO_2$ observations for $NH_3$ emission quantification, because the two are intrinsically connected. The method reduces our dependence on accurate prior emissions of either $SO_2$ or $NH_3$. It also allows to correct the $SO_2$ effect, without rerun the model with corrected $SO_2$ emissions. So it should be easier to implement than the approach described by the referee.

One line 186, you mention that 'we perform GEOS-Chem full chemistry simulations in selected years'. What was the GEOS-Chem mode run initially for the full 2008-2018 period? This should be moved to Section 2.2 and expanded on, i.e. give a brief description of the simulation modes and their difference, why you chose to use the full chemistry mode for only these three years, etc... Also, with the way the paper is currently organized, it is a bit confusing figuring out which simulation mode was used for what. I assume that the non-full-chemistry mode was used with Eqns. (1) and (3), but I can't see where this is specified in the paper. Please clarify in Section 2.2 exactly what the output of each simulation mode is used for. A table might make this clearer to the reader.

Sorry for the confused organization. All the GEOS-Chem simulations we run are in the full chemistry mode. We have corrected it as the 'GEOS-Chem simulation'.

In regards to the calculation of fractional bias of IASI as compared to GEOS-Chem, if the IASI observations were already used in the estimates of the emissions through comparisons to GEOS-Chem (Eqns. (1) and (3)), what is the goal of presenting these FB values? This comparison uses the full chemistry mode (I assumed that Eqns. (1) and (3) where used in non-full chemistry mode, is that correct?), but I'm not clear on what this comparison is telling you. Add more explanation of what this is being used for. If comparisons with an independent observation data set are added, as suggested above, is this comparison still necessary?

We now add more clarification in the text (**Line 332-338**). This exercise is to check whether results from Eq. (1) and (3) (which are a simplified description of $NH_3$ chemistry and transport) indeed improve agreements with IASI.

On line 263, it says that the 'large post-2013 increase is inconsistent with flat or even declining fertilizer input and manure amount (Fig. 4).' Do the plots in Fig. 4 include emissions from livestock? If not, it may be hard to really tell if the trends found by using Eqn. (1) are really inconsistent with the data in Fig. 4.

We now clarify that livestock emissions are related to livestock manure amount (**Fig. 4 and Line 265**).

On line 275, it described that $SO_2$ trends from OMI and OMPS are used with Eqn (3). The estimates of $SO_2$ trends from OMI and OMPS will have uncertainties associated

with them. Could these uncertainties be incorporated into your analysis? i.e. could you include another sensitivity test for the value of the $SO_2$ trend used?

We report the suggested sensitivity test in **Line171**. Yes, Wang and Wang (2020) reported the stand deviation of the yearly $SO_2$ trend values as 10-20 %. Thus, we perturb the $\omega$ in the Eq. (3) as 4/6 % and -4/-6 % over China and India in our uncertainty analysis (**Line 187-188**).

**Comment #3: Minor comments**

Line 31: 'particulate matters' to 'particulate matter'

Line 33: 'These particulate matters also affect' to 'Particulate matter also affects'
Line 35: '... and wet processes, and therefore are associated with ...' remove the word 'therefore'

Line 50: 'depending' should be changed to 'dependent'

Line 136: Would be good to remind readers here that $C_{\{NH3,obs\}}$ are the monthly mean observations, not an individual observation.

Line 180: Replace 'related to the number of IASI measurements (ð• ' › ) and their measurement errors:' by 'given by'

Line 187: 'top-down estimate (TDE) and prior emissions (BUE1)' the acronym TDE was already defined on line 134, while the acronym BUE was used prior to this line but not defined until this line. Change so that these acronyms are defined at their first appearance in the paper.

Section 3.2 starts by introducing Fig. 2 and then introduces Fig. 3 in the next sentence. Try to reorganize this section by first discussing Fig. 2 and then move on to discussing Fig. 3 instead of introducing them at the start of the section together.

We correct them, thank you for the careful read.

In regards to Fig. 2, I'm a bit confused about the colour bars and the colour scale. Fig 2c shows the difference between Fig 2a and 2b, so the units should be in g m^{-2} a^{-1}, but Fig 2c has blue in it, which isn't in the colour bar in the top row that has these units. The bottom row colour bar has blue in it, but it has the wrong units for 2c (g m^{-2} a^{-1} per decade). Could you clarify which colour bar goes with Fig 2c? Could the figure be rearrange or changed in some way to make this more clear?

The units for **Fig. 2c** are g m$^{-2}$ a$^{-1}$ and we add a color bar for **Fig. 2c.**

[Figure]

**Figure 2.** Spatial distribution of NH3 emission fluxes during 2008-2018. (a) Bottom-up emissions (BUE1), (b) top-down emissions (TDE) inferred from IASI observations, (c) difference between TDE and BUE1 estimates and (d) emission trends derived from TDE estimates. Green boxes denote seven regions analyzed in Sect. 3.2. Top-down emission fluxes are computed with Eq. (1) except for IP and EC where Eq. (3) is applied. Linear trends are computed from the time series of annual averages. Dots in (d) represent significant linear trends at the 95 % confidence level.

In the description of Fig 3, it says 'Shadings represent the range derived from uncertainty analyses' Table S1 describes a number of different sensitivity tests. Does the shading refer to the upper and lower bounds over all tests? Please specify in the main text.

We add the description of the uncertainty analysis as the referee suggested (**Line 185-195**)

Line 210: 'Both the satellite and model do not find significant trends in $NH_3$ concentrations over India (absolute value less than 1 % $yr^{-1}$).' Looking at Figs. 1(c) and (d), it looks like the trend over India are not insignificant. Could you confirm this?

Thank you for your suggestion. The overall observed $NH_3$ concentration trend over India are -0.8 % $a^{-1}$ (p = 0.25) while trends in south-eastern part are significantly decreasing ($\sim$ -3.8 % $a^{-1}$, p < 0.05). The model does not find significant trends in $NH_3$ concentrations, with p > 0.1 in all grids. We revised the text accordingly (**Line 215-217**).

[Figure]

**Figure A.** Spatial distribution of relative linear trend of (a) IASI and (b) GEOS-Chem NH$_3$ column concentrations in India. Dots indicate that linear trends are significant at the 95 % confidence levels (p value < 0.05). Linear trends are computed from the time series of annual averages.

Line 261: Why is Eqn. (1) 'based only on NH3 column measurements' if Eqn. (1) blends information from the observations with a priori information of the emissions?

We correct following the suggestion (**Line 270**)

Line 295: Is the lifetime of 21.2 +/- 3.8 h the lifetime of NH3 (as stated) or is it the lifetime of NHx?

It is the lifetime of NH$_3$ as stated.

In Table S1, 'TDE' isn't really a 'parameter perturbed'. This line should probably be something like 'None' or 'None (TDE)'.

We correct **Table 1** following the suggestion.

**3. RESPONSES TO REFEREE, REFEREE #2 (Minor Revision)**

**Comment #1: In regards to the comparison with ground-based observations:**

    a. Could you add a line or two to very briefly describe each observation data set (e.g. instrument type, active or passive sampler, integration time, observation frequency, etc…)?

    We add the text in **Line 345-352**.

b. For the comparison of the RB and RMSE stats between BUE1 and TDE, could you quantify the statistical significance of these differences? Without uncertainty estimates on these stats, its hard to tell if, for example, the difference in FB values for JFM are meaningful or not.

We add the Mann-Whitney U test for the comparison between surface measurements with simulation driven by BUE1 and TDE (**Line 363**).

c. On lines 342-347, you discuss the limited impact of the top-down estimate on the comparison with the surface observations and say one possible explanation is 'systematic differences between satellite and surface measurements'. Could the weighting in Eq. (1) and/or (3) also be a cause of this? i.e. If you increasing the weighting of the IASI data would the FB values move closer to zero (at least for JFM and AMJ where TDE shows a slight improvement in FB over BUE1)? If this is not the case and the top-down emissions are being moved to mean values away from those near the surface observations because there is an underlying bias between the surface observations and IASI, could you quantify the magnitude of this bias?

Our consistency evaluation (Fig. S5) shows that simulations driven by either BUE1 or TDE show relatively small fractional biases (<0.5) in regions where we have these surface observations (i.e., North America, Europe, and south-eastern Asia), suggesting that our estimates are already in line with the IASI data. It is unlikely that further pushing the estimates towards IASI data can greatly improve agreement with surface observations.

However, we acknowledge that there may be reasons other than systematic observation biases (or inconsistency as total column and surface measurements are not directly comparable) to explain this discrepancy. In general, it is not straightforward to evaluate whether total column and surface measurements are consistent. We now remove the sentence to avoid the impression that we are sure this being the major reason. This information was added into the section labeled 'Uncertainty evaluation'. I'm not sure this title describes this information. Probably better to put this information into its own subsection.

We add a new subsection titled "Comparison to independent surface networks" (**Line 344**)

**Comment #2: Lines 129-134: Thank you for adding this information. However, I think some of these sentences need to be clarified.**

a. 'We note that the ANNI-NH3-v3R retrieval does not provide averaging kernels (Whitburn et al., 2016; Van Damme et al., 2021).' Looking at Whitburn et al. 2016, the neural network based retrieval algorithm does not use an averaging kernel to make the retrieval. Is this correct? If so, from the wording of this sentence it sounds like averaging kernels were used but not included in the data product. Could you clarify this point?

b. Sentences in lines 130 and 131 begin with 'However' and 'Besides', which link these sentences together. But I'm a bit confused why these sentences are being linked together based on their content. Could you reword or further clarify this?

We have revised the sentences to avoid the confusion (**Line 129-132**).

**Comment #3&4:**

a. Line 325: You write on this line 'ranges of perturbation tests divided by their averages'. Are these ranges then divided by 2? i.e. (upper-lower)/2?

We revised the relevant sentences to improve clarity (**Line 327**).

b. Fig. 7: It might be good to add a plot of the absolute uncertainty (units of Tg a-1 per area or similar) so you can better discern what's going on in regions with relative uncertainty bigger than 100%. If these regions all have small absolute uncertainty, then we can easily see that the large relative uncertainties were cause by small average values. But there are quite a few regions with the relative uncertainty larger than 100%, so it would be good to know if this is the case with all of these regions or if there are regions with non-negligible average values and large uncertainties.

Thank you for your suggestion. We add a panel in **Fig.7** to show absolute uncertainty derived from the perturbation tests and briefly discuss the results in the main text.

**Comment #5: Thanks for adding the sensitivity tests for the SO2 trend values. I see the description in the lines you mentioned in your reply, but I can't see where the results of this sensitivity test is. I might have just missed this in the text, in which case could you direct me to these lines, and if not could you add some text about these results.**

Thanks for pointing this out. We now report the results of this sensitivity test in **Line 286-287**.

**REFERENCE**

Evangeliou, N., Balkanski, Y., Eckhardt, S., Cozic, A., Van Damme, M., Coheur, P.-F., Clarisse, L., Shephard, M. W., Cady-Pereira, K. E., and Hauglustaine, D.: 10-year satellite-constrained fluxes of ammonia improve performance of chemistry transport models, Atmospheric Chemistry and Physics, 21, 4431-4451, 10.5194/acp-21-4431-2021, 2021.

Van Damme, M., Whitburn, S., Clarisse, L., Clerbaux, C., Hurtmans, D., and Coheur, P.-F.: Version 2 of the IASI NH3; neural network retrieval algorithm: near-real-time

and reanalysed datasets, Atmospheric Measurement Techniques, 10, 4905-4914, 10.5194/amt-10-4905-2017, 2017.

Van Damme, M., Clarisse, L., Whitburn, S., Hadji-Lazaro, J., Hurtmans, D., Clerbaux, C., and Coheur, P. F.: Industrial and agricultural ammonia point sources exposed, Nature, 564, 99-103, 10.1038/s41586-018-0747-1, 2018.

van der Werf, G. R., Randerson, J. T., Giglio, L., van Leeuwen, T. T., Chen, Y., Rogers, B. M., Mu, M., van Marle, M. J. E., Morton, D. C., Collatz, G. J., Yokelson, R. J., and Kasibhatla, P. S.: Global fire emissions estimates during 1997–2016, Earth System Science Data, 9, 697-720, 10.5194/essd-9-697-2017, 2017.

Wang, Y. and Wang, J.: Tropospheric SO2 and NO2 in 2012–2018: Contrasting views of two sensors (OMI and OMPS) from space, Atmospheric Environment, 223, 10.1016/j.atmosenv.2019.117214, 2020.